# D-ark—A Shared Digital Performance Art Archive with a Modular Metadata Schema

**Anni Saisto** [1,*,†] and **T.E.H.D.A.S.** [2,†]

1  Pori Art Museum, Eteläranta, 28130 Pori, Finland
2  Radioasemantie, 28330 Pori, Finland; dada@tehdasry.fi
*  Correspondence: anni.saisto@pori.fi; Tel.: +358-44-7011083
†  These authors contributed equally to this work.

**Abstract:** Digital objects and documentation of intangible cultural heritage pose new challenges for most museums, which have a long history in preserving tangible objects. Art museums, however, have been working with digital objects for some decades, as they have been collecting media art. Yet, performance art as an ephemeral art form has been a challenge for art museums' collection work. This article presents a method for archiving digital and audiovisual performance documentation. D-ark (digital performance art archive) is based on a joint effort by the artist community T.E.H.D.A.S., which has created the archive, and Pori Art Museum, which is committed to preserving the archive for the future. The aim is to produce sufficient standardized metadata to support this objective. This article addresses the problems of documenting an ephemeral art form and copyright issues pertaining to both the artist and the videographer. The concept of D-ark includes a modular metadata schema that makes a distinction between descriptive, administrative, and technical metadata. The model is designed to be flexible—new modules of objects or technical metadata can be added in the future, if necessary. D-ark metadata schema deploys the FRBRoo, Premis, VideoMD, and AudioMD standards. Administrative and technical metadata modules abide by Finnish digital preservation specifications.

**Keywords:** performance art; archiving; digital preservation; metadata schema

## 1. Introduction

How does one archive a local artist community's performance art video documentation? How does one produce high-quality metadata and fulfil the requirements of Finnish digital preservation? Digital objects and documentation of intangible cultural heritage pose new challenges for most museums, which have a long history in preserving tangible objects. Art museums, however, have been working with digital objects for several decades, as they have been collecting media art. Yet, performance art as a time-based ephemeral art form offers a challenge to traditional art museum practices.

The problems of collecting and documenting performance art have been surveyed in the academic field from various points of view—in the United Kingdom, the Theatre and Performance Research Association (TaPRA) has been studying performance documentation processes in working groups since 2005 [1]. Zürcher Hochschule der Künste (Zurich University of the Arts) carried out the project "archiv performativ: A model concept for the documentation and reactivation of performance art" from 2010–2012 [2]; it focused on the relationship between performance documentation and transcription. The University of Oxford/Ruskin School of Art, the University of Girona, and the artist in residence program GlogaurAIR in Berlin have been collaborating in the creation of the European Live Art Archive (ELAA) to share information about networks, knowledge, and documentation [3]; at the moment, the archive contains interviews with the artists. The Tate museum carried out a research

project to examine practices for collecting and conserving performance art from 2012–2014. As a result, Dutch and British academic scholars and museum professionals created the Live List. It is a practical and thorough set of prompts to consider when a live work enters a collection [4].

Universities or other organizations own several performance archives. Perhaps one of the oldest is Franklin Furnace Archive Inc., located in New York. It has been preserving time-based art, including performance, since 1976 [5]. The archive contains videos, images, press releases, and printed material such as posters. The database can be accessed on site. Die Schwarze Lade—The Black Kit—an archive of international performance art, was founded in 1981 by approximately 70 West German and European artists [6]. The Black kit is maintained by ASA-European (Art Service Association) and is seen as part of a living network of artists and ideas. A vast archive includes documents, art relics, photographs, and films about the performing arts, performance art, and performative interventions. Currently, the archive is being indexed and catalogued so that a digital database can be created. The University of Bristol maintains the Live Art Archives, established in 1994. The archive includes over 200,000 records of live art, and the university claims it to be the largest record in the world [7]. One part of the Live Art Archives is a digital archive, which can be accessed via the internet as a PDF with short descriptions of the recordings [8]. In the field of contemporary dance, research has been done about the documentation of performing arts and three-dimensional movement. The Inside Movement Knowledge research project concentrated on the documentation, transmission, and preservation of contemporary dance and choreographies. The project was led by a consortium of Amsterdam school of Arts, International Choreographic Arts Center ICK, Netherlands Media Art Institute, and University of Utrecht. [9]. Siobhan Davies Replay is a vast choreographic archive on the internet that documents the work of Siobhan Davies Dance Studios. The descriptive metadata is enriched with photographs, posters, newspaper articles, etc. [10]. Motionbank is an archive created for choreographic practice. The project has been carried out by The Forsythe Company since 2010 [11]. The recent publication *Histories of Performance Documentation: Museum, Artistic, and Scholarly Practices* (Routledge, London, New York, 2018) addresses the ways in which art museums have approached performance works. The book includes interviews with museum professionals around the world explaining how they have presented, collected, and archived performance art. The question about whether performance documentation is treated as a work of art belonging to a collection or as a documentation that should be archived is discussed, and different points of views are represented. Associate curator Eric Crosby from the Walker Art Center of Minneapolis states that museums should consider whether the performance work is an object or an event; choosing the latter option would be a difficult shift for collecting institutions to make [12]. The CIDOC (Icom International Committee for Documentation) Exhibition and Performance Documentation Working Group, founded in 2015 [13], aims to map the current situation of performance documentation and archiving practices in the museums.

During the development of D-ark (digital performance art archive), we searched for examples of metadata schemas for performance work but were unable to find examples suitable for our needs—a schema in which performance is an immaterial work, an event to which digital files are connected. Thus, this article presents the results of the development work, a method to operate with digital performance video documentation in an art museum context. Called D-ark ("digital performance art archive"), it was created to preserve video documentation produced by the artist community T.E.H.D.A.S., situated in Pori, Finland. This article will shortly describe the collaboration model, problems of archiving performance art, copyright matters, and Finnish digital preservation requirements, as well explain the metadata schema and its potential extensions. A selection of D-ark videos is available online at http://www.tehdasry.fi/dark/. At the moment, D-ark contains 300 video recordings, 61 of which are published on the internet. The online database for collecting the metadata is available to authorized users.

The D-ark metadata schema is modular—the core is a description of the performance. Other parts are administrative information about the video documentation and technical metadata concerning the video file. The idea behind the modular metadata schema is flexibility—it enables additions, e.g., residual objects or textual interpretations, as well as the addition of new technical metadata if the video file has to be migrated to a new file format. The schema has been created using FRBRoo (A Conceptual Model for Bibliographic Information in Object-Oriented Formalism) [14] and standards recommended in the Finnish digital long-term preservation specifications [15].

## 2. Partners and Origins of the Project

T.E.H.D.A.S. is an artist collective founded in 2002 and situated in the city of Pori on the west coast of Finland. T.E.H.D.A.S. currently has 113 members, and as a group, it functions as a democratic collective. As a group, they emphasize anonymity yet afford each member freedom for personal artistic development. Their mission is to offer a wide range of art experiences that move between different cultural fields. The recipient of the State Art Award in 2011, the collective has become famous for performance festivals and clubs arranged in Pori and Tampere. They have actively organized performance events since 2005 and have systematically recorded the performances on video. Over the years, T.E.H.D.A.S. has developed a significant archive of over 300 recordings containing Finnish and international performance art. The archive was originally developed by Antti Pedrozo, Eero Yli-Vakkuri, and Manu Alakarhu, and it included a method for gathering descriptive metadata.

Pori Art Museum is a municipal art museum run by the City of Pori. It is focused on contemporary art and modernism. Pori Art Museum was founded in 1981 and in its exhibitions and events has surveyed ephemeral art such as Fluxus, environmental art, and site-specific art. As an international contemporary art museum, it has arranged solo shows by artists such as Richard Long (1986), Daniel Buren (1988), Gordon Matta-Clark (1987), Yoko Ono (1991), Dennis Oppenheim (1993), Geoffrey Hendricks (1995), Jimmie Durham (1997), and Fred Sandback (2011). The museum is responsible for documenting visual culture in its area. Since 2013, T.E.H.D.A.S. and Pori Art Museum have collaborated to develop a video archive tailored for performance art. The original idea of a joint effort came from Esko Nummelin, the director of Pori Art Museum. The model was developed in two projects funded by the Ministry of Education and Culture and the National Board of Antiquities in the years 2013 and 2014, respectively. The joint mission is to archive performance art documentation produced by T.E.H.D.A.S. and to enable its preservation for the future. The museum is committed to preserving the archive and to provide it to the national digital long-term preservation system. Members of T.E.H.D.A.S. serve as curators of the archive, and Pori Art Museum supports the process with its expertise on archiving. As a result, the archive D-ark was born. It consists of three parts—digital storage of master files, a simple database for descriptive and technical metadata of the recordings, and an online archive on the website http://www.tehdasry.fi/dark/.

Archivist Juha Mehtäläinen from T.E.H.D.A.S. has played a key role as an organizer of the archive. He has defined the metadata for describing performances and prepared the material for digital preservation. Antti Pedrozo, an artist and member of T.E.H.D.A.S., created the IT-infrastructure, including the D-ark website. Chief curator Anni Saisto from Pori Art Museum first mapped the descriptive metadata using CIDOC-CRM, then the FRBRoo standard. She created a modular metadata schema, including technical metadata required by Finnish digital preservation specifications. There is still work to be done cataloguing and preparing the material for the national digital preservation service.

### 3. Problems of Archiving Performance Art Documentation

The role of documentation in relation to works of performance art has been reflected on from several different aspects. Philip Auslander has created two categories for performance art documentation—documentary and theatrical. The first category contains the idea of a document as evidence of a performance that occurred in a specific time and place. The second category, theatrical, includes performance art staged for the camera—it has not existed as an independent artwork, nor was it performed for an audience [16]. According to how T.E.H.D.A.S. sees their documentation, their work falls into the first category. Thus, performance is seen as a singular moment during which the artwork exists. The documentation is proof of what happened, a glimpse of a moment that has since disappeared into the past. This thinking has led us to the conclusion that a video recording and metadata describing the performance is documentation, not an independent artwork authorized by the artist. For this reason, T.E.H.D.A.S. performance art documentation is considered in creating an archive, not a collection as it is understood in the art museum context.

Auslander sees documentation itself as performative—the act of documenting something as a performance is what constitutes it as a work of art. The established practice in performance events is that a camera focuses on the performer instead of interaction between the artist and the audience. As a result, the camera frames the performance and participants in the tradition of reproducing artworks instead of documenting events in an ethnographic way [16]. T.E.H.D.A.S. has aimed to avoid disturbing the performer or the audience while documenting, resulting in that the camera is often in the background or to the side, therefore allowing the audience to be seen in the picture as well. The footage does not exclude the audience, yet the main focus remains on the performance. As well, the framing depends on the nature of the performance and possible interaction between the artist and the audience. When considered from this point of view, one can say that T.E.H.D.A.S. has documented events rather than reproduced artworks. Yet, as Daniela Salazar reminds us, the person or people recording the performance always make an interpretation of what is taking place. The recording is somebody's reinterpretation executed with available technical means [17].

The attitudes towards performance art documentation have long been critical. Toni Sant reminds us that emphasizing the dichotomy between the event and its recording might no longer be relevant. He continues that professionally executed archiving supports the artists, who can then acquire different perspectives on their body of work, as can researchers and the general public, for whom documentation provides information about an ephemeral art form and its development over the years [18]. In his recent book *Reactivations*, Philip Auslander points out that "documentation is a present act directed to a future audience" [19]. For T.E.H.D.A.S., the archive serves as a memory that carries information about the association's past activities. Its other aim is to promote performance art, both by making it available for public via the internet and by gathering sufficient metadata for researchers. D-ark has gained appreciation among artists as their work is being documented and published on the website. It is also possible that the era of smart phones and social media have changed the attitudes towards documentation. Video recording has become a mundane activity carried out by ordinary people. Especially younger generations consider it to be a natural part of life, as well as of performance clubs and events.

## 4. Ownership and Copyright

When addressing the issue of copyright, three questions arise—the copyright of the artist, the copyright of the videographer, and the role of the art museum in relation to an archive owned by T.E.H.D.A.S. These questions need to be taken into consideration when drawing up contracts. Firstly, I will discuss the artist's copyright. When organizing performance events, T.E.H.D.A.S. has drawn up contracts with artists, and the copyright questions of the video recording have been taken into consideration. The content of the contracts has varied over the years. The basic idea has been that, when agreeing on performing in an event arranged by T.E.H.D.A.S., the artists have consented that the organizer can record their performances. Contracts give T.E.H.D.A.S. the right to use the recordings to promote performance art in the association's own activities and to archive it. The other option is that artists can give their consent to T.E.H.D.A.S. for publishing the recordings on the internet. The D-ark website (http://www.tehdasry.fi/dark/) was created for that purpose. In this way, the copyright could be held by both the publisher and the artist.

Secondly, when a performance artwork is recorded, a copyright is also born to the videographer. According to Finnish law, the copyright exists for 50 years from when the recording was made. Members of T.E.H.D.A.S. have been responsible for recording, and they have agreed to pass on their rights to the artist association. According to the law, the name of the videographer has to be mentioned when the videos are being shown. The third question is the position of Pori Art Museum in relation to an archive owned by T.E.H.D.A.S. At the moment, T.E.H.D.A.S. is the owner of the archive, and the archival files are deposited in Pori Art Museum. Among the rights that the artist grants in the contract to T.E.H.D.A.S. is that the documentation can be moved to a third party if it is a memory institution collaborating with T.E.H.D.A.S. If the artist association were to be dissolved in the future, the archive can be transferred to Pori Art Museum to secure its existence.

To survey the problems of copyright and possibilities of Creative Commons licensing in the art field, Pori Art Museum arranged a seminar on 8 December 2017. It was curated by Juha Mehtäläinen, and the speakers of the seminar were artists experienced in both using licensed material in their artistic work and publishing their own works on the internet using free licenses. The presentations and discussions demonstrated how multifaceted the questions concerning copyright and the freedom of internet can be. Both positive and negative experiences and opinions were shared, identifying the benefits of especially the CC0 license and the attention one's work can earn on the internet. On the other hand, the power of big corporations and algorithms directing people's orientation on the internet were criticized. A Finnish-speaking web publication, "Luova yhteismaa ja taide" ("Creative Commons and Art"), addressed issues discussed at the seminar [20].

## 5. Digital Preservation Guidelines in Finland

The Ministry of Culture and Education of Finland launched a project in 2008 to ensure the high quality of digital cultural heritage information. One aim of the National Digital Library project was to develop a long-term preservation service for Finnish libraries, archives, and museums [21]. Digital preservation guidelines and IT infrastructure were developed by CSC–IT Center for Science Ltd. Metadata describing the informational content, provenance information, and directions regarding the usage of the content were considered to have a key role. The digital preservation service was introduced in 2015 [22]. The National Digital Library project ended in 2017, and by that time, material from seven organizations was transferred to the digital preservation service [23]. From now on, the service is being run by CSC–IT Center for Science Ltd.

The National Digital Library Project compiled specifications for producing sufficient metadata for long-term preservation. Specifications are metadata requirements and preparing content for digital preservation, file formats, digital preservation service interfaces, and schema catalog and schematron rules [24]. The specifications are quite technical and detailed. Distilling the essential guidelines from a vast quantity of material can be challenging for professionals working with cultural heritage processes in smaller museums. In 2014, project researcher Sakari Hanhimäki and chief curator Anni

Saisto studied the specifications and how to apply them to video files. Using recommended file types, verifying the data integrity through use of message digest algorithms, and gathering a large amount of technical metadata are main elements to be taken into consideration. As the next step, the metadata should be encoded and wrapped using Metadata Encoding & Transmission Standard (METS). This part is technical and will be addressed when data can be transferred to the digital long-term service.

Digital preservation service supports 20 different kinds of metadata standards. One of them, LIDO, can be used to map the metadata described with FRBRoo. According to digital preservation specifications, video is among the most complicated file type, as it must be described with three different standards—Premis for the birth history of the digital object, copyright status, and general technical information; VideoMD for picture; and AudioMD for sound. Due to this fact, the amount of technical metadata is large, up to 50 pieces of information. A large part of metadata can be gathered from the file info, yet some bits of the information should be written down when the files are being created and edited. For bigger organizations, it may be possible to produce the metadata efficiently, as they have technical personnel and large amounts of similar digital objects. Smaller museums face challenges though, as they have limited resources and heterogeneous material with variable birth histories.

## 6. Modular Metadata Schema and Application of FRBRoo

The metadata schema created for D-ark consists of three modules, and four different standards have been applied (Figure 1). The modules contain information about the performance (FRBRoo), administrative metadata of the documentation (Premis), and technical metadata explaining the qualities of the digital objects (PREMIS, VideoMD, AudioMD). This model distinguishes between recorded performance event and the recording. For instance, the duration of the performance may be different from the duration of the video recording, the first piece of information belonging to the performance metadata and the latter to technical metadata. As well, the possibility of confusing different authors—the author of the performance and the author of the video recording—has been eliminated.

FRBRoo is a conceptual model that provides a unified conceptual model for bibliographic records. It allows the modeling of creative processes in a coherent and integrated way. FRBRoo inherited features from both CIDOC CRM (Cidoc Conceptual Reference Model) and FRBR (Functional Requirements for Bibliographic Records) and it can be used for describing performing arts [25]. Alberto Pendón Martínez and Gema Bueno de la Fuente have discussed potential description models for documenting performance. They recommend using FRBRoo, as it provides a suitable conceptual basis compatible and interoperable with other institutions [26]. The mapped metadata for D-ark performance art documentation consists of three main elements that are also the core of FRBRoo—F20 performance work, F25 performance plan, and F31 performance. Performance work is the intellectual concept of the artwork, performance plan is a blueprint for how to realize the performance work, and performance is the event at which the art work is being presented.

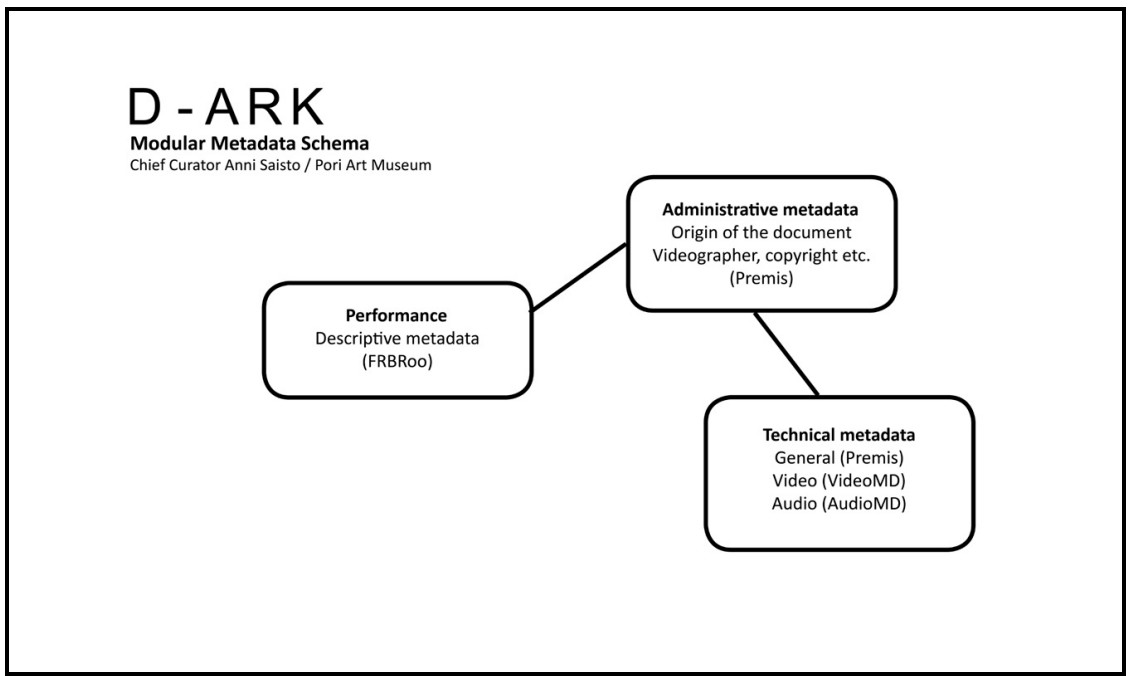

**Figure 1.** The modular metadata schema of D-ark.

In the D-ark metadata schema (Figure 2), performance work can have descriptions or keywords (E62 String, E55 Type) and a title (E75 Conceptual Object Appellation). The T.E.H.D.A.S. archive does not include written or drawn plans made by the artist (F25 Performance Plan), yet it is included in the mapping, as it is the semantic glue joining together performance work and the activity resulting from it, the F31 performance. In D-ark, F31 performance has the largest number of qualities connected to it—the performer (E39 Actor) who can have a real name (E82 Actor Appellation), qualities such as an artist (E55 Type) and a nationality (E74 Group.) The performance happens somewhere (F9 Place) with an additional definition (E44 Place Appellation). The performer can use specific object/s (E70 Things) and the performance has happened at a certain time (E50 date) and also has a duration (E52 Time-span). The performance is documented on a video (E31 Document) which has administrative and technical metadata (E62 String) described with Premis, Video MD, and Audio MD standards. This mapping can be adjusted or enlarged when needed. For instance, it contains the possibility to add derivative works in the future. As FRBRoo understands the complexity of performance art consisting of an abstract idea and its different manifestations, it is the most suitable metadata format for describing performance artworks.

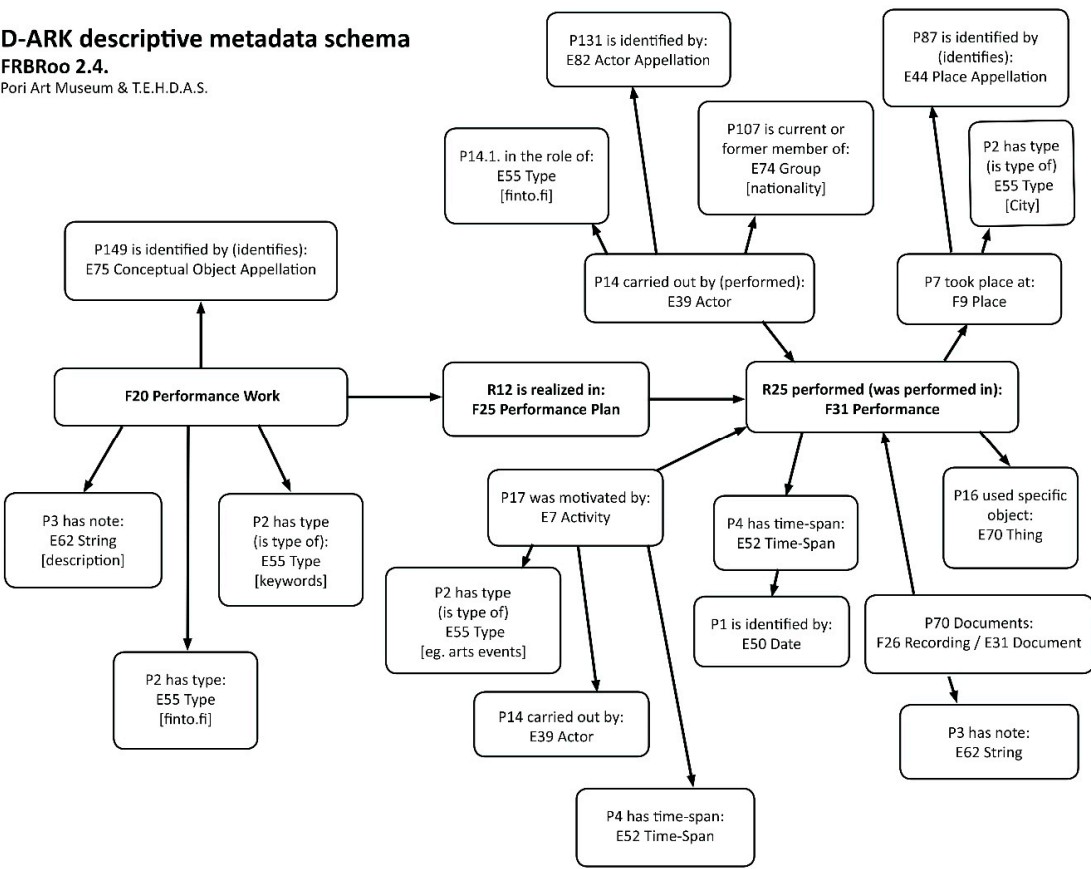

**Figure 2.** FRBRoo (A Conceptual Model for Bibliographic Information in Object-Oriented Formalism) metadata schema for D-ark.

## 7. Administrative and Technical Metadata

The administrative metadata is described with PREMIS Data Dictionary for Preservation Metadata maintained by the Network Development and MARC Standards Office of the Library of Congress. PREMIS is an international standard for supporting the preservation of digital objects and their long-term usability [27]. The current version is 3.0. In the D-ark metadata schema, PREMIS is used to describe the documentation event, the copyright status of the digital object (including the country whose laws apply), and the determination date of the copyright status. It is also used to indicate some information about the digital object, such as the message digest algorithm and the software used to create it.

In the D-ark metadata schema, PREMIS is also used to indicate the actions the preservation repository is allowed to take. PREMIS version 2.2. proposed a controlled vocabulary, such as to replicate (make an exact copy), migrate (make a copy identical in content in a different file format), modify (make a version different in content), use (read without copying or modifying), disseminate (create a copy or version for use outside of the preservation repository), and delete (remove from the repository) [28]. The vocabulary is useful in the context of long-term preservation, as permission for possible migration to a new file format in the future can be articulated clearly. Finnish digital preservation specifications list file formats that can be either transferred or retained in the repository. Even though file formats would be retainable, according to present guidelines the migration might, in the future, be indispensable to secure the usability of digital objects.

To provide sufficient information for digital preservation, video files are described with VideoMD and AudioMD standards. Like PREMIS, they are maintained by the Library of Congress Network Development and MARC Standards Office. The development of standards is being undertaken by the larger METS community and other experts in the field [29]. This data can be treated as an object characteristics extension in PREMIS. In the metadata schema of D-ark, features such as duration, bit rate, color, codec and resolution information, display aspect ratio, and signal format are being collected. As the videos have sound, a separate AudioMD extension is being added that includes, e.g., audio data encoding, codec information, sampling frequency, and number of audio channels.

## 8. Possible Extensions of the Modular Metadata Schema

The idea behind the modular metadata schema is flexibility—it can be expanded with new elements in the future (Figure 3). For instance, if digital video files need to be migrated into a new file format to secure their preservation, a new module of technical metadata can be added in connection with the original video file data. Information about the old technical metadata is retained so that the original nature of the digital file can be traced. The other possibility is to add completely new objects, either tangible or digital. If T.E.H.D.A.S. wants to expand the archive in the future—for instance with texts, photographs, or residual objects from the performance—it is possible to add new modules to the schema.

If digital texts and photographs are added, both could include at least three modules: Firstly, descriptive metadata about the content of the text or the photograph; secondly, administrative metadata indicating the copyright status and birth history of the document; and, thirdly, technical metadata. As with the video documentation, the performance remains the topic of the documentation, so the modules are connected to it. Thus, the benefit of the modular metadata schema is that the performance needs to be described only once. Naturally, suitable standards should be applied when creating new modules, and the recommendations of digital preservation specifications should be taken into consideration.

When expanding the modular metadata schema, the metadata of residual objects differ from digital files. First, physical objects have to be described. This part of the process resembles traditional museum work and cataloguing objects. The objects have been part of the performance in some way, so they are connected to the performance module. If objects are photographed, the photographs most likely will not require separate descriptions about their visual content. Thus, the administrative metadata describing the copyright status and birth history of the photograph can be directly connected to the residual object module. Digital photography naturally has its own technical metadata, which creates the third module for this object type. Following this logic, the modular metadata schema can be expanded with new object types, yet the capacity and qualities of the collections management system define the possibilities of new additions.

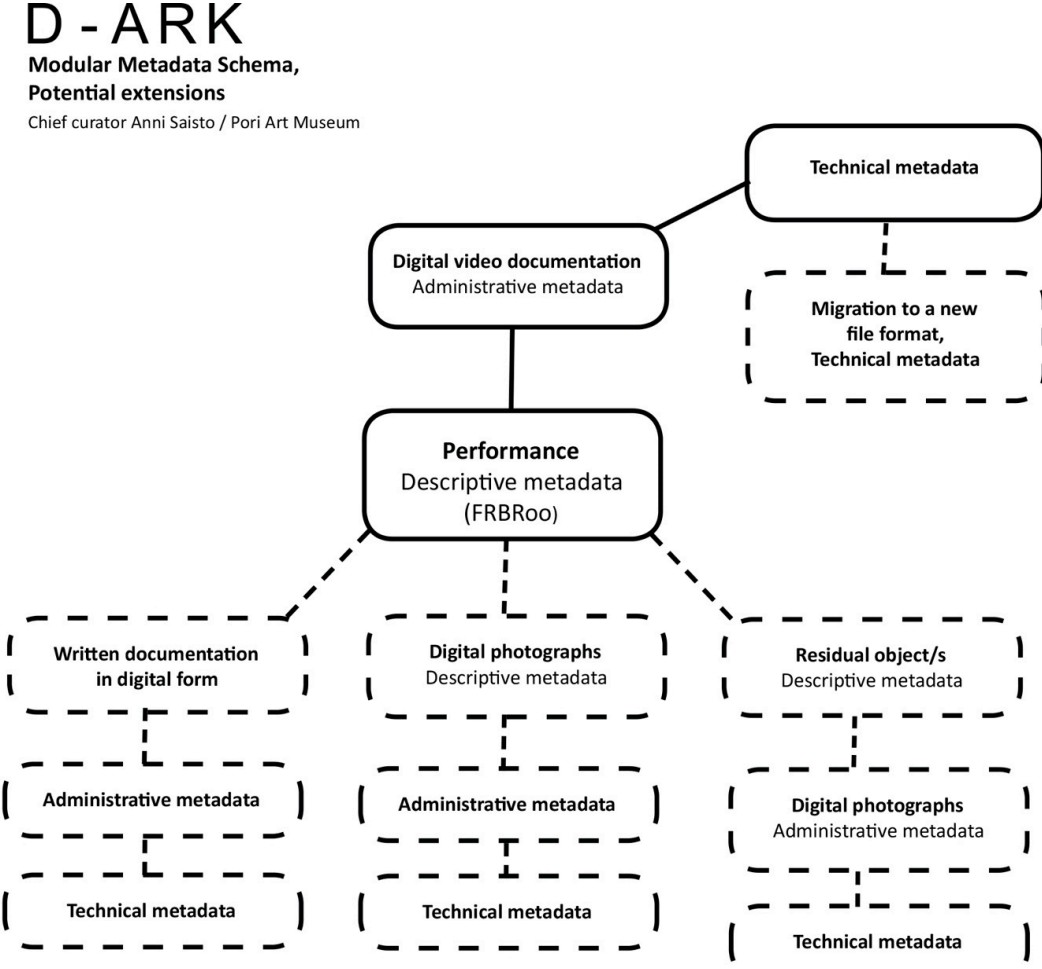

**Figure 3.** Potential extensions of the modular metadata schema.

## 9. Conclusion and Future Challenges

The process of developing archiving methods with T.E.H.D.A.S. started in 2013. So far, D-ark performance archive has been established and organized, copyright issues have been addressed, and metadata schemas have been created. For the creators of D-ark, it has been a rewarding journey crisscrossing live art, archiving, and the digital world, and creating new connections between the three. A lot remains to be done. Cataloguing and preparing the archive for the digital preservation service requires a good deal of work. The metadata schema is in the process of publication and will be made available in English on the D-ark website at http://www.tehdasry.fi/dark/ by the end of June 2019. The archive will be transferred to the national digital preservation repository when the service becomes available to smaller institutions. There will be a lot to learn for everybody working with D-ark.

Perhaps the biggest challenge in meeting the requirements of digital preservation is the updating of specifications. Understandably, such a thing is necessary, as the world of digital technology evolves constantly, but it might not be possible to collect the lacking metadata retrospectively if digital objects are transferred to the preservation service years after the metadata was gathered. It is also worth noting that even though metadata standards are being updated, a digital preservation service might not support the newest versions. However, Finnish digital preservation specifications offer abundant and thoroughly constructed guidelines on how to work with digital objects, which file types to favor, and how to apply international standards and vocabularies. For a museum employee with a

background in humanities, this poses a challenge; nevertheless, efforts are rewarded as a new world of digital archiving opens up.

In all, D-ark is a never-ending learning process that has only just begun. The study of the fields of performance art, archiving and digital world develop constantly. The relationship between the performance and its documentation is being reflected on and reinterpreted. Digital archiving practices make progress constantly, and archiving meets new demands as the European Union creates new legislation to protect its citizens' personal data and copyrights in the digital era. In the future, the D-ark website presenting a selection of video documentation could be developed further by providing, for instance, a possibility to share the audience's experiences and thoughts about performances. The archive could also be expanded by adding photographs and other types of material containing information about the performance. These expansions depend on resources such as funding and people's time and motivation. It would also require an advanced collections management system.

The shared ownership and the responsibilities present a challenge as well. The continued existence and development of the archive are dependent on the community that has created it, for its members have a personal relationship with the material as well as with the archiving methods designed to support it. If people leave or are replaced, there is a risk that the archive will be left aside and its development will halt. As in every archive, the material is alive when it is being used. If not, the archive falls dormant or ceases to exist, as it is no longer meaningful to anybody. The metadata can be exported from D-ark database in XML and CSV formats. This enables further use of the data, for instance in the Wikidata context, as connecting the data to existing object-oriented collections management systems might be challenging. When actively used by artists, researchers, and other audiences, D-ark fulfils its purpose—helping people to remember, become acquainted with, and reinterpret performance art. Perhaps it could also offer some ideas for the documentation processes of intangible cultural heritage.

**Author Contributions:** Conceptualization, A.S. and T.E.H.D.A.S.; funding acquisition, A.S.; methodology, A.S. and T.E.H.D.A.S.; project administration, A.S.; visualization, A.S. and T.E.H.D.A.S.; writing—original draft, A.S.; writing—review and editing, A.S. and T.E.H.D.A.S.

**Funding:** This research was funded by the opetus- ja kulttuuriministeriö (Ministry of Education and Culture) and Museovirasto (The Finnish Heritage Agency), Finland.

**Conflicts of Interest:** The authors declare no conflict of interest. The funders had no role in the design of the study; in the collection, analyses, or interpretation of data; in the writing of the manuscript, or in the decision to publish the results.

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
