# Peer review of "D-ark—A Shared Digital Performance Art Archive with a Modular Metadata Schema"

_heritage, doi:10.3390/heritage2010064_

Round 1

Reviewer 1 Report

This is an interesting piece - I think the literature review section could be condensed (most debates are well known) and the original structure expanded. It's worth looking at the Tate Live list.

Author Response

Response to Reviewer 1 Comments:

Point 1:

I think the literature review section could be condensed (most debates are well known) and the original structure expanded.

Response 1:
I condensed the literature review in chapter 2 and expanded the original structure.

Point 2:

It's worth looking at the Tate Live list.

Response 2:

I added the Tate Live list to the introduction.

Reviewer 2 Report

In general the article presents a good overview of past and existing practices around the documentation and archival possibilties  of performance art. Obvious ommissions are  perhaps  the research project Collecting the Performative (RCE, Van Abbemuseum and Tate), and other research that  could have been of interest here is taking place in the closely related field of contemporary dance (see, among others, Inside Movement Knowledge, Shioban Davies' work in archiving, or Motionbank). Yet what is lacking in this overview is in what ways this model, D-ark, is contributing to the existent models that are around.

The author clearly outlines the main challenges around documenting performance art and it points to the issues around copyright (although an additional remark could be made about sound in case that is used). Yet it remains a bit unclear what and how to further the discussion. The explanation of D-ark is also very clear and the model seems useful in particular  in its core descriptions and also in its adaptability over time. What is less convincing is the emphasis on administrative and particularly technical metadata. While the discussion on metadata is thriving particularly in archival circles, at the moment there's a lot of critique on the emphasis of this type of digital (technical) metadata, mostly because the metadata can often be retrieved automatically or that certain metadata information is rather meaningless, for instance information about a specific  codec as these become quickly obsolete. Hence the question of these types of metadata is becoming less relevant. While it could be argued that the discussion is ongoing it is a shame that the author doesn't take these considerations, or counter arguments, into account. In that sense, the article remains rather descriptive of the D-ark model, and is not very critical of its implementation and the necessity of all the information that needs to be provided. Another important question that remains unanswered is how to insert this data (of the D-ark) into, or connect it to, existing registration or data management models that a museum might be using. The usability and sustainability of the model will not only depend on the people connected to it (which as the author rightly outlines is a key challenge), but mostly to the adaptability to existing systems and workflows.

Author Response

Response to Reviewer 2 Comments

Point 1: Obvious ommissions are  perhaps  the research project Collecting the Performative (RCE, Van Abbemuseum and Tate)

Response 1: I added the Collecting the Performative project on the introduction.

Point 2: ther research that  could have been of interest here is taking place in the closely related field of contemporary dance (see, among others, Inside Movement Knowledge, Shioban Davies' work in archiving, or Motionbank).

Response 2: I added these examples to the introduction.

Point 3: Yet what is lacking in this overview is in what ways this model, D-ark, is contributing to the existent models that are around.

Response 3: I have not been able to find actual metadata schemas that would have been used for describing performance art and its documentation. The material I have been able to find I have presented in the introduction.

Point 4: What is less convincing is the emphasis on administrative and particularly technical metadata. While the discussion on metadata is thriving particularly in archival circles, at the moment there's a lot of critique on the emphasis of this type of digital (technical) metadata, mostly because the metadata can often be retrieved automatically or that certain metadata information is rather meaningless, for instance information about a specific  codec as these become quickly obsolete. Hence the question of these types of metadata is becoming less relevant. While it could be argued that the discussion is ongoing it is a shame that the author doesn't take these considerations, or counter arguments, into account.

Response 4: As an art museum curator I have concentrated on creating the descriptive metadata schema with FRBRoo and how to connect it with administrative and technical metadata. The technical metadata has been included following the guidelines given by Finnish digital preservation specialists, and this is a field a cannot estimate from a critical, academic point of view. Yet there are some bits of information that cannot be retrieved from the file info, such as the software and its version number that was used for editing the file.

Point 6: Another important question that remains unanswered is how to insert this data (of the D-ark) into, or connect it to, existing registration or data management models that a museum might be using. The usability and sustainability of the model will not only depend on the people connected to it (which as the author rightly outlines is a key challenge), but mostly to the adaptability to existing systems and workflows.

Response 6: I added a point to the conclusion that explains the possibilities of interoperable data.

Reviewer 3 Report

Digital preservation is very updated not only for cultural institutions but nowadays to all organisations. When we talk about digital art we also have a very diverse number of contentes, but we also have digital art itself. Storage of digital art is a very importante matter and not only of associated documents or images.

Cataloguing and classyfying information requires a global work. Institutions working this matters should create a common project at European level to achieve a common framework.  It isn't to define some of the new concepts being developped and to understand the future importance of some of them, so this isn't na esay task. The article shows a relevant work.

Author Response

Response to Reviewer 3 Comments

Point 1: I was unable to find any comments that would require revision.